# FiLM: Fill-in Language Models for Any-Order Generation

## Abstract

Language models have become the backbone of today's AI systems. However, their predominant left-to-right generation limits the use of bidirectional context, which is essential for tasks that involve filling text in the middle. We propose the **F**ill-**in** **L**anguage **M**odel (FiLM), a new language modeling approach that allows for flexible generation at any position without adhering to a specific generation order. Its training extends the masked language modeling objective by adopting varying mask probabilities sampled from the Beta distribution to enhance the generative capabilities of FiLM. During inference, FiLM can seamlessly insert missing phrases, sentences, or paragraphs, ensuring that the outputs are fluent and are coherent with the surrounding context. In both automatic and human evaluations, FiLM outperforms existing infilling methods that rely on left-to-right language models trained on rearranged text segments. FiLM is easy to implement and can be either trained from scratch or fine-tuned from a left-to-right language model. Notably, as the model size grows, FiLM's perplexity approaches that of strong left-to-right language models of similar sizes, indicating FiLM's scalability and potential as a large language model.[1]

## 1 Introduction

Large language models (LLMs) have demonstrated remarkable success in open-ended text generation and a variety of natural language understanding and reasoning tasks (Brown et al., 2020; Wei et al., 2022; Ouyang et al., 2022). The next word prediction objective, inherent in the training of these models, has positioned them as predominantly Causal Language Models (CLMs). However, this confines their generation order to left-to-right, constraining their versatility and applicability, particularly in tasks that require filling in the middle (Zhu et al., 2019).

We present the **F**ill-**in** **L**anguage **M**odel (FiLM), designed for flexible sequence generation in any desired order. As depicted in Fig. 1, FiLM exhibits the capability to fill in text segments at any specified position, taking into account both the preceding and subsequent context. This distinctive attribute opens up avenues for a myriad of applications, including but not limited to, assisting in text editing and revision, automating form filling, and facilitating code completion.

Training FiLM takes insights from both Masked Language Models (MLMs) (Devlin et al., 2018) and text diffusion models (Li et al., 2022; Austin et al., 2021; Dieleman et al., 2022). Unlike MLMs that are trained with a fixed mask ratio, FiLM adopts a strategy inspired by diffusion models that utilize varying noise levels (§3.1). For each training sequence, a mask probability is drawn from the Beta distribution, and each token is masked with this probability. The model then learns to predict the masked tokens based on the surrounding context. This adaptive masking strategy significantly enhances FiLM's generative capacity (§4.1).

At decoding time, FiLM has the flexibility to start with either a sequence entirely of masks or a partially complete text interspersed with masks. It progressively replaces one mask with a predicted token at each step. We explore various decoding orders for FiLM (§3.2), and our analysis shows that, besides proceeding from the leftmost mask to the right, selecting the mask position with the minimum entropy is also an effective strategy (§4.1).

---

[1]Our code will be publicly released after the review period.

They met again only once, in 745. In 746, he moved to the capital in an attempt to resurrect his official career. He took the civil service exam a second time during the following year, but all the candidates were failed by the prime minister ( apparently in order to prevent the emergence of possible rivals ). He never again attempted the examinations, instead petitioning the emperor directly in 751, 752, and 753. After 752, he is recorded as being promoted to the position of chief of the guards in the Chang 'an Palace. It is unclear whether he also received a post as an official in the capital. In 754, he was appointed as a major in the central government, although only because of the massive military buildup at the time. It was in that year that Du Fu was forced to move his family due to the turmoil of a famine brought about by massive floods in the region. In 755, he received an appointment as Registrar of the Right Commandant's office of the Crown Prince's Palace. Although this post was not very prestigious in normal times it would have been at least the start of an official career. Even before he had begun work, however, the position was swept away by events. = = = War = = = The An Lushan Rebellion began in December 755, and was not completely suppressed for almost eight years. It caused enormous disruption to Chinese society : the census of 754 recorded 52 @. @ 9 million people, but ten years later, the census counted just 16 @. @ 9 million, the remainder having been displaced. In addition, some 3 @. @ 6 million people were killed in the rebellion. By 758, it had killed an estimated 2 @. @ 7 million people. The later Chinese historian Sima Qian records that famine and civil strife had killed as many as 25 million people in 757 alone. Official records from the time give a total of 142 million people. Although the catastrophe was not unprecedented, it was a stark sign that the Chinese government could not deal with the level of the disaster. Even the emperor was taken aback by the scale of the devastation. When the news reached him on December 27, 757, he wrote : This is the greatest calamity in which I have lived through, if even I know such suffering, the common man must surely be rattled by the winds. In the end, Emperor Xuanzong was forced to flee the capital and abdicate.

Figure 1: Flexible sequence infilling by FiLM-1.6B. The given context is in black, and the text generated by the model is in color.

FiLM can be either trained from scratch or fine-tuned from off-the-shelf MLMs or CLMs. In this paper, we experiment with the latter setting, which avoids the expensive pretraining stage and is appealing in practice. We develop a method to evaluate the perplexity of any-order language models, enabling a direct comparison between FiLM and CLM (§3.3). When fine-tuned from GPT2-xl, FiLM yields perplexity of $14.03$ and $20.32$ on the WikiText-103 and One Billion Word datasets, respectively. While these numbers lag behind those by CLM fine-tuned from GPT2-xl on the same data ($11.29$ and $16.46$), we observe a diminishing disparity with an increasing model size. Specifically, as the pretrained model scales up from GPT2-small to GPT2-xl, the gap narrows from $5.85$ to $2.74$ and from $7.96$ to $3.86$ on the respective datasets. This trend indicates FiLM's promising potential with further scaling up, positing it as a viable alternative in the realm of LLMs (§4.2).

FiLM excels in filling text in the middle and substantially outperforms previous state-of-the-art infilling methods that employ a CLM trained on rearranged data (Donahue et al., 2020; Fried et al., 2022; Bavarian et al., 2022). In particular, in our human evaluation for story completion, FiLM is favored in 48% of the cases over a specially trained CLM that is four times larger, while the latter is preferred in only 21%, with the remaining cases resulting in ties.

## 2 RELATED WORK

FiLM can be viewed as an extension of MLMs, specifically tailored to enhance generative capabilities. While MLMs utilizing bidirectional context have shown exceptional performance in language understanding tasks (Devlin et al., 2018; Liu et al., 2019), their application in generation has been limited. A line of work represented by T5 (Raffel et al., 2020) and BART (Lewis et al., 2019) employs an MLM-style encoder but remains dependent on a CLM decoder. Another approach attempts to transform MLMs into generators by interpreting them as Markov random fields or energy-based models and using Markov-chain Monte-Carlo (MCMC) sampling algorithms for decoding (Wang & Cho, 2019; Goyal et al., 2021). Despite the intricacy of these techniques, the generative performance of such models still lags significantly behind that of CLM. Distinctively, FiLM is a decoder-only model that leverages bidirectional context and achieves superior text infilling performance.

Previous studies have investigated conditional MLMs for non-autoregressive machine translation (Ghazvininejad et al., 2019), with the goal of speeding up generation by decoding multiple tokens per step (Gu et al., 2017; Stern et al., 2019). FiLM also possesses the ability to simultaneously fill in multiple masks in each iteration. Nonetheless, in the absence of the anchoring provided by source sentences in translation tasks, tokens often exhibit strong interdependence. Making independent predictions in such scenarios could compromise the quality of the generated text. Recognizing this challenge, this paper adopts a sequential decoding approach, filling in one mask at a time conditioned on previous predictions. The exploration of strategies to accelerate generation with FiLM is a promising direction for future research (Chen et al., 2023).

Apart from FiLM, several other any-order language models possess the ability to generate text in a non left-to-right order, notably including XLNet (Yang et al., 2019), the General Language Model

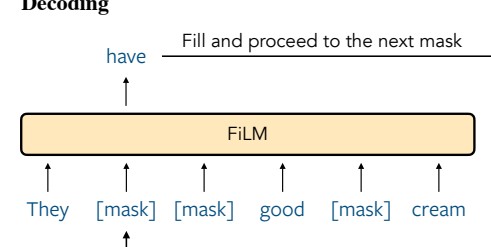

Figure 2: Training and decoding of FiLM. During training, the mask probability $p$ is sampled according to a noise schedule, and then each token is independently replaced with `[MASK]` with probability $p$; FiLM is trained to predict the original tokens at the masked positions. At decoding time, the masks are sequentially filled in, each conditioned on the given context and previous predictions.

(GLM; Du et al. 2021), and the Blank Language Model (BLM; Shen et al. 2020). XLNet employs a permutation language modeling objective for training, while GLM adopts an autoregressive blank infilling objective. However, these models have primarily been developed for language understanding tasks and generating text from scratch, with a limited focus on text infilling performance. BLM is explicitly designed for filling in blanks, but it faces the challenge of markedly higher perplexity compared to CLM, which is a significant disadvantage. FiLM achieves competitive perplexity with CLM, demonstrating its potential as a versatile tool for text generation.

## 3 FILL-IN LANGUAGE MODELS (FILM)

FiLM uses a special `[MASK]` token to indicate positions to be filled. It can operate in two modes: (1) generate text from scratch by populating a sequence consisting entirely of masks; (2) start from partial text and fill in the masked positions. In this section, we dive in to the training of FiLM, and how to decode from it. Additionally, we extend the established perplexity evaluation for any-order language models, enabling a direct comparison between FiLM and Causal Language Models (CLMs).

### 3.1 TRAINING

Given a training sequence $x$ consisting of tokens $(x_1, \ldots, x_n)$, we first sample the mask probability $p$ according to a noise schedule, then independently mask each token $x_i$ with probability $p$. Let $\tilde{x} = (\tilde{x}_1, \ldots, \tilde{x}_n)$ denote the resulting masked sequence. FiLM takes $\tilde{x}$ as input and is trained to predict the original tokens in $x$ at the masked positions of $\tilde{x}$, as illustrated in Fig. 2 (Left). The training process is designed so that FiLM leverages both the left and right contextual information available for each mask position when making predictions.

An intuitive initial choice for the noise schedule is sampling $p$ from the uniform distribution $U[0, 1]$ (Liao et al., 2020). This approach ensures that FiLM learns to generate text from sequences with varying numbers of masks, which is crucial for generation from scratch. However, assigning equal weights across different mask probabilities can be suboptimal, as a high mask ratio leaves the model with insufficient information for predictions, while a low mask ratio oversimplifies the task.

To address this, we turn to the beta distribution, defined over the interval $[0, 1]$ and characterized by two shape parameters, $\alpha$ and $\beta$. Different values of $\alpha, \beta$ allow for skewing towards different values of mask probabilities and thereby avoiding extreme values. Specifically, considering that the mode of $\text{Beta}(\alpha, \beta)$ is $\frac{\alpha-1}{\alpha+\beta-2}$ for $\alpha, \beta > 1$, we maintain a constant sum of $\alpha + \beta$ (set to 5 based on favorable results from our experiments), and adjust them to produce modes between $0.1$ and $0.9$ at intervals of $0.1$. The distributions obtained are depicted in Fig. 3 (Left). We empirically compare the performance of FiLM trained using different noise schedules: with $p$ sampled from the uniform distribution or from the beta distribution with varying modes, to determine the most effective strategy.

Note that when $p$ is fixed, we recover the Masked Language Modeling (MLM) objective. For instance, BERT employs a mask ratio of $0.15$ (Devlin et al., 2018). Although training with a fixed mask ratio

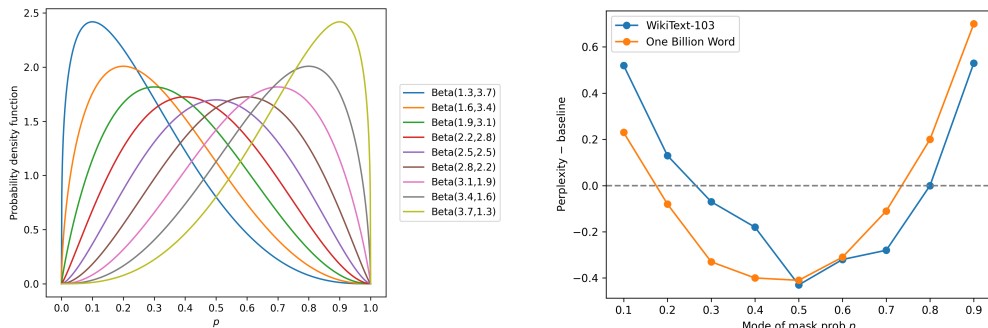

Figure 3: Left: Illustration of the Beta distribution with varying modes. Right: Perplexity of FiLM when trained with different Beta distributions, each value depicted as the deviation from the baseline perplexity achieve by the uniform distribution.

can be effective for representation learning, as we shall see in §4.1, it significantly hinders the model's capacity to generate text from scratch.

## 3.2 DECODING

At decoding time, given a sequence $\tilde{x}$ of incomplete text that contains masks, FiLM fills in one mask at each step, conditioning on the provided context and previous predictions. This process is iterated until no masks remain. Fig. 2 (Right) illustrates this procedure.

When there are multiple masks in $\tilde{x}$, FiLM needs to determine which mask to fill in first. Two straightforward strategies are: (1) making a random selection, which reflects FiLM's training process; (2) generating in a unidirectional manner, either from the leftmost mask to the right or vice versa. This mirrors a CLM, but with the additional conditioning on the subsequent context.

In addition, we explore two adaptive strategies based on the probability distribution predicted by the model for each mask: (3) selecting the mask position with the minimum entropy, indicating the model's highest certainty; (4) selecting the position with the maximum entropy, where the model is least certain. The min-entropy strategy acts as a heuristic to search for an "easy-first" decoding order, whereas max-entropy pursues a "hard-first" order. Note that the generation order in these approaches is not predetermined but is established step by step during the model's decoding process.

Upon determining the decoding order, conventional decoding algorithms of CLM—such as sampling, greedy decoding, and beam search—are equally compatible with FiLM. In our experiments, we use sampling and evaluate the performance of FiLM under different decoding orders. We will observe in §4.1 that left-to-right and min-entropy decoding strategies consistently perform well, whereas random and max-entropy decoding strategies are less effective.

## 3.3 PERPLEXITY

To calculate the perplexity of FiLM, we need to compute the probability it assigns to a sequence. Although marginalizing over all possible sequence generation orders is intractable, evaluating FiLM with a specific decoding order is feasible. Our method of computing perplexity can be used to evaluate other any-order language models as well.

Specifically, we first compute $p_{\text{len}}(n)$ by calculating the frequency of sequence length $n$ in the training data[2]. Subsequently, we can determine the log-probability of generating a sequence $x = (x_1, \ldots, x_n)$ with a decoding order $\sigma$, where $\sigma$ is an $n$-permutation. Note that $\sigma$ can be deterministic, such as left-to-right and right-to-left; random; or adaptive, like the min-entropy and max-entropy strategies discussed in the previous subsection.

---

[2]We apply add-one smoothing to $p_{\text{len}}(n)$ to avoid assigning a probability of zero to unseen sequence lengths.

Let $\theta$ represent the model parameters. We define $p_\theta(x_{\sigma_t}|x_{\sigma_1}, \ldots, x_{\sigma_{t-1}}, n)$ as the probability that FiLM predicts $x_{\sigma_t}$ given an $n$-length sequence with $x_{\sigma_1}, \ldots, x_{\sigma_{t-1}}$ filled. For instance, consider $n = 4$, $\sigma = (3, 1, 4, 2)$, and $t = 3$, $p_\theta(x_4|x_3, x_1, 4)$ is the probability of predicting $x_4$ at the last mask position in the sequence $(x_1, \texttt{[MASK]}, x_3, \texttt{[MASK]})$. We have:

$$\log p_\theta(x; \sigma) = \log p_{\text{len}}(n) + \sum_{t=1}^{n} \log p_\theta(x_{\sigma_t}|x_{\sigma_1}, \ldots, x_{\sigma_{t-1}}, n) \tag{1}$$

The perplexity is then computed as $\exp\left(-\frac{1}{n+1}\log p_\theta(x; \sigma)\right)$. We divide by $n + 1$ to ensure comparability with CLM, which appends an $\texttt{[EOS]}$ token to $(x_1, \ldots, x_n)$ to signify the end of generation, resulting in a total sequence length of $n + 1$. In contrast, FiLM determines the sequence length through $p_{\text{len}}(n)$ and then fills in an $n$-length sequence.

This calculated perplexity not only serves as a metric for comparison with CLM but also for evaluating the effectiveness of FiLM's various training and decoding strategies to identify the optimal approach.

## 4    Experiments

In this section, we first empirically analyze FiLM's various training and decoding strategies, as discussed in §3.1 and §3.2, to identify the optimal configuration (§4.1). Subsequently, we compare FiLM with CLM on language modeling (§4.2). Lastly, we test FiLM for text infilling (§4.3) and story completion (§4.4) and design evaluation protocols to compare it with previous infilling methods in terms of fluency, coherence, and logical consistency with the surrounding context.

**Datasets**    We conduct experiments on three datasets: WikiText-103 (WT-103; Merity et al., 2016), One Billion Word (1BW; Chelba et al., 2013), and ROCStories (Mostafazadeh et al., 2016).

WikiText-103 is a collection of Wikipedia articles with 103M words in total. Each article has several thousand words, which we chunk into windows of 512 tokens. One Billion Word is a sentence-level dataset, with an average length of 28.5 tokens and a total of 1B words. ROCStories consists of five-sentence commonsense stories, each averaging 51.4 tokens in length, totaling 5M words.

**Experimental setup**    We evaluate the performance of FiLM when fine-tuned from both an MLM and a CLM. For this investigation, we employ two pretrained models: RoBERTa (Liu et al., 2019), representing MLMs, and GPT2 (Radford et al., 2019), representing CLMs. Note that the causal masking should be deactivated when fine-tuning FiLM from a CLM.

RoBERTa is available in two sizes: base (124M parameters) and large (355M), while GPT2 is offered in four sizes: small (124M), medium (355M), large (774M), and xl (1558M). The models are trained using the Adam optimizer with a learning rate of $2\mathrm{e}-5$ and a batch size of 20K tokens. Training is conducted for 500K steps on WikiText-103 and One Billion Word datasets, and for 50K steps on ROCStories. Utilizing automatic mixed precision (AMP), our largest model based on GPT2-xl takes about one week to train using two 80G A100 GPUs.

### 4.1    Analysis of FiLM

In the following analysis, we compare the perplexity of FiLM (§3.3) under various training noise schedules (§3.1) and decoding orders (§3.2) to determine the most effective strategy. We use pretrained models RoBERTa-base and GPT2-small here for efficiency, and report results on the validation sets of WikiText-103 and One Billion Word.

We first examine the impact of employing different noise schedules for training FiLM. We use the left-to-right decoding here, with an in-depth investigation into decoding orders to follow. Table 1 shows that a fixed noise schedule $\delta(0.15)$ leads to substantially higher perplexity on both datasets, highlighting the necessity for a variable mask probability $p$ in order to enhance generative capacity. To elucidate the effects of training with the Beta distribution featuring varying modes, we plot the difference in perplexity compared to $U[0, 1]$ in Fig. 3 (Right). The findings corroborate our hypothesis that overly small or large values for the mode of $p$ are suboptimal. The lowest perplexity is achieved at $\text{Beta}(2.5, 2.5)$ with mode 0.5, marking an improvement of approximately 0.4 over that of $U[0, 1]$. In light of these results, we adopt the $\text{Beta}(2.5, 2.5)$ noise schedule in subsequent experiments.

Table 1: Perplexity of FiLM under different training noise schedules.

| Dataset | Pretrained model | $\delta(0.15)$ | $U[0,1]$ | Beta$(2.5, 2.5)$ |
|---------|------------------|----------------|----------|-------------------|
| WT-103 | RoBERTa-base | 26.76 | 19.02 | **18.59** |
| 1BW | RoBERTa-base | 34.68 | 30.56 | **30.15** |

Table 2: Perplexity of FiLM using different decoding orders. The **bold** numbers highlight the best perplexity, while the underlined ones denote the second best.

| Dataset | Pretrained model | Random | L2R | R2L | Min-Ent | Max-Ent |
|---------|------------------|--------|-----|-----|---------|---------|
| WT-103 | RoBERTa-base | 19.95 | **18.59** | 19.04 | 18.70 | 21.38 |
|         | GPT2-small | 28.42 | **21.68** | 29.34 | 23.47 | 30.68 |
| 1BW | RoBERTa-base | 31.85 | 30.15 | **29.92** | 30.07 | 33.16 |
|     | GPT2-small | 41.52 | **32.55** | 42.73 | 34.59 | 44.04 |

Next, we investigate the effects of decoding from FiLM in different orders. The results are presented in Table 2. While FiLM is trained to predict a random subset of words, decoding from left to right substantially outperforms decoding in a random order. This finding is consistent with the inherent sequential nature of language. Intriguingly, when fine-tuned from an order-agnostic MLM, FiLM demonstrates near-optimal performance with right-to-left decoding, even attaining the lowest perplexity on One Billion Word. However, when fine-tuned from a left-to-right CLM, the efficacy of right-to-left decoding significantly degrades, lagging behind even random decoding. This illustrates the resistance in altering a CLM's behavior from left-to-right to right-to-left.

The min-entropy order consistently emerges as the second-best strategy, while the max-entropy order proves to be the least effective. Fig. 4 showcases the decoding process for both min-entropy and max-entropy orders. The min-entropy strategy generates text in a segmented manner, sequentially predicting cohesive phrases such as "thank you" and "for your service", deferring the more uncertain name after "Mr." to the final stage. In contrast, the max-entropy strategy opts for a "challenging" order, selecting distant positions at each step.

Given the simplicity and superior performance of left-to-right decoding, irrespective of whether FiLM is fine-tuned from an MLM or a CLM, we select this order for FiLM in subsequent experiments.

## 4.2 LANGUAGE MODELING

In this subsection, we evaluate the performance of FiLM on language modeling and compare it with CLM. Both FiLM and CLM are fine-tuned on WikiText-103 and One Billion Word, using pretrained models of varying sizes. The resulting perplexities are plotted in Fig. 5.

The choice of the pretrained model has a significant influence on the performance of FiLM. When fine-tuned from RoBERTa, which incorporates bidirectional context during pretraining, FiLM demon-

| Min-entropy | Max-entropy |
|-------------|-------------|
| [M] [M] [M] [M] [M] [M] [M] [M] [M] [M] | [M] [M] [M] [M] [M] [M] [M] [M] [M] [M] |
| [M] [M] [M] [M] [M] [M] [M] [M] [M] . | [M] [M] [M] [M] [M] [M] [M] [M] service [M] |
| Mr [M] [M] [M] [M] [M] [M] [M] [M] . | [M] . [M] [M] [M] [M] [M] [M] service [M] |
| Mr . [M] [M] [M] [M] [M] [M] [M] . | [M] . [M] [M] thank [M] [M] [M] service [M] |
| Mr . [M] [M] thank [M] [M] [M] [M] . | [M] . Chairman [M] thank [M] [M] [M] service [M] |
| Mr . [M] [M] thank you [M] [M] [M] . | Mr . Chairman [M] thank [M] [M] [M] service [M] |
| Mr . [M] , thank you [M] [M] [M] . | Mr . Chairman [M] thank [M] [M] your service [M] |
| Mr . [M] , thank you for [M] [M] . | Mr . Chairman , thank [M] [M] your service [M] |
| Mr . [M] , thank you for your [M] . | Mr . Chairman , thank [M] [M] your service . |
| Mr . [M] , thank you for your service . | Mr . Chairman , thank you [M] your service . |
| Mr . Chairman , thank you for your service . | Mr . Chairman , thank you for your service . |

Figure 4: An illustration of FiLM decoded using min-entropy and max-entropy orders. The mask position selected to be filled in at each step is highlighted in the green color.

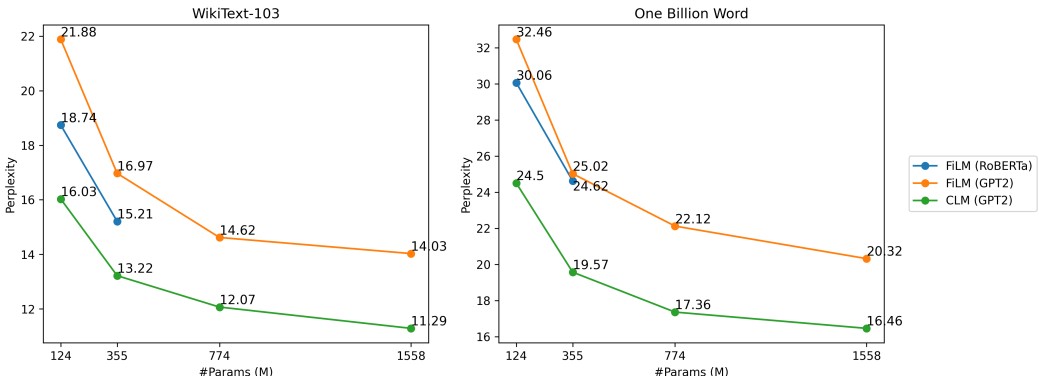

Figure 5: Perplexity of FiLM and CLM on WikiText-103 (Left) and One Billion Word (Right).

strates superior performance compared to when it is fine-tuned from GPT2, which leverages only unidirectional context during pretraining.

While FiLM exhibits higher perplexity than CLM when generating text from scratch, this gap diminishes as the model size increases. As the pretrained model scales from 124M GPT2-small to 1558M GPT2-xl, the difference in perplexity decreases from 5.85 to 2.74 on WikiText-103 and from 7.96 to 3.86 on One Billion Word. This narrowing gap suggests that FiLM could benefit from further scaling and holds considerable potential as an alternative LLM.

## 4.3 TEXT INFILLING

We evaluate FiLM's text infilling performance using the WikiText-103 and One Billion Word datasets. For a given sequence $x$ of length $n$, we first sample the number of spans $m$ from 1 to 5. Subsequently, we draw $2m$ numbers from 1 to $n$ without replacement and sort them to get $a_1, \ldots, a_{2m}$ as the endpoints of each span. Tokens in $x$ located between $[a_{2i-1}, a_{2i})$ $(i = 1, \ldots, m)$ are masked, and the model is tasked with filling in these spans. Given that WikiText-103 comprises long documents, the infilling tasks on this dataset involve composing multiple sentences. In contrast, the One Billion Word dataset consists of individual sentences, and the infilling tasks are primarily at the phrase level.

Previous state-of-the-art methods for infilling have predominantly relied on training CLMs on rearranged data (Donahue et al., 2020; Aghajanyan et al., 2022; Fried et al., 2022; Bavarian et al., 2022). In these methods, random spans of text are replaced with special sentinel tokens and moved to the end of the sequence. Then a CLM is trained to generate text in this modified order. For instance, the manipulated sequence for the example in Fig. 2 would appear as "They [MASK:0] good [MASK:1] cream [FILL:0] have really [FILL:1] ice". At test time, the tokens generated after the given context "They [MASK:0] good [MASK:1] cream [FILL:0]" are re-integrated into the corresponding mask positions. This approach is referred to as causal masking (CM). We fine-tune both FiLM and CM from GPT2-xl and use top-p sampling with a threshold of 0.95 and a temperature of 0.8 for decoding.

To evaluate the model outputs, we compute the ROUGE scores (Lin, 2004) against the original text to measure their overlap. Specifically, ROUGE-1, ROUGE-2, and ROUGE-L measure the overlap of unigrams, bigrams, and the longest common subsequence between the generated and reference texts, respectively. Since there may be valid infillings different from the original, we also employ GPT4 for evaluation (Liu et al., 2023). We present the outputs generated by FiLM and CM to GPT4 in a random order, and instruct it to determine which option is more grammatically fluent and coherent with the surrounding context. When neither option is more fitting than the other, GPT4 is directed to declare a tie. This GPT4 evaluation is conducted on 500 examples from the test set.

As shown in Fig. 6, FiLM demonstrates a significant advantage over CM, improving the average ROUGE score by 10.12 on WikiText-103 and 7.35 and One Billion Word. Moreover, GPT4 prefers the outputs by FiLM over CM with margins of 8.4% and 5.4% on the respective datasets. Fig. 7 displays example infillings, where GPT4 accurately identifies repetitions and inconsistencies generated by

| Dataset | Model | ROUGE-1/2/L |
|---------|-------|-------------|
| WT-103 | CM-1.6B | 21.73/7.14/16.66 |
|         | FiLM-1.6B | **35.34/13.29/27.26** |
| 1BW | CM-1.6B | 33.69/8.88/33.54 |
|     | FiLM-1.6B | **42.55/13.35/42.25** |

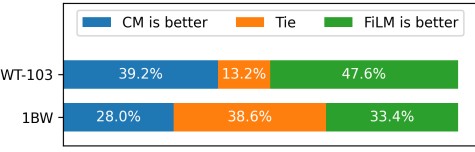

Figure 6: Text infilling results on WikiText-103 and One Billion Word. The left table presents the ROUGE scores, and the right figure illustrates the comparative evaluation by GPT4.

| | |
|---|---|
| 1. CM | Depression, stress and stress increase the risk of alcoholism, gambling, excessive drinking and drug abuse. |
| 2. FiLM | Depression, loneliness and stress increase the risk of, say, drug abuse. |
| GPT4 | Option 1 is grammatically correct, but it contains redundancy by stating that stress increases the risk of excessive drinking and alcoholism, which essentially means the same thing. Option 2 is also grammatically correct, but includes 'say' which adds unnecessary hesitation to the sentence. Nonetheless, option 2 resolves the redundancy issue present in option 1, making it the better completion for the given sentence. For those reasons, the final decision is: <2>. |
| 2. CM | In the same way as the Himalayas are " Asia's water-tower, " Switzerland is the source of Europe's biggest rivers, supporting agriculture and farming, and a world leader in nuclear power stations. |
| 1. FiLM | In the same way as the Himalayas are " Asia's water-tower, " Switzerland is the source of Europe's biggest rivers, supporting agriculture and the construction of new nuclear power stations. |
| GPT4 | Option 1 is superior in terms of grammatical fluency and coherence. It expresses that the water from Switzerland's rivers supports both agriculture and the construction of new nuclear power stations, which makes clear and logical sense. Option 2, by contrast, seems a bit disjointed. It indicates that the water supports agriculture, farming (a redundancy since farming is a type of agriculture), and then asserts that Switzerland is a world leader in nuclear power stations. The conjunction "and" makes it seem as if being a world leader in nuclear power stations is another thing that the water is supporting, which isn't logically coherent. The final decision is <1>. |
| 1. CM | The situation has been further complicated by growing indications that the US Treasury is preparing to inject capital directly into banks. |
| 2. FiLM | The situation has been further complicated by growing indications that the US Treasury is preparing to sell stakes in some of America's banks. |
| GPT4 | Both options are grammatically correct. However, they suggest different outcomes. In option 1, the US treasury is preparing to support the banks by injecting capital directly into them. This implies a strategy of financially bolstering banks. In option 2, the US treasury is preparing to divest its stake in some banks, which implies a different strategy of reducing government involvement or exposure in the banking sector. Both fit the context and maintain the coherence of the sentence. Hence, the choice between the two hinges on the intended meaning rather than on grammar or coherence. Therefore, the answer is <->. |

Figure 7: Text infilling examples from One Billion Word, evaluated by GPT4. The provided context is in black, and the model generated text is in color. The number before each model denotes the option number presented to GPT4 for evaluation.

CM. Despite CM's attempt to consider subsequent context by artificially altering the text order, it still introduces redundancy by inserting "stress" between "Depression," and "and stress", and commits a logical error by adding "farming, and a world leader in" between "Switzerland is the source of Europe's biggest rivers, supporting agriculture and" and "nuclear power stations". In contrast, FiLM generates apt fillings such as "loneliness" and "the construction of new", resulting in coherent sentences. Additional examples are available in Fig. 10 and Fig. 11 in the Appendix.

### 4.4 COMMONSENSE STORY COMPLETION

In this set of experiments, we assess FiLM's proficiency in logically completing commonsense stories using the ROCStories dataset. For each story, we randomly remove one of the five sentences and ask the model to fill it in. In addition to ROUGE scores and GPT4 evaluation, we conduct human evaluation to determine which model output best preserves the story's logical flow and coherence. We present 100 examples to human judges and collect two labels for each. Under the three categories "Option 1 is better", "Option 2 is better", and "Tie", GPT4 and human evaluators reach consensus $49.5\%$ of the time, while the agreement between humans is $62\%$. Table 3 in the Appendix provides further details of the agreement analysis.

Due to the relatively small size of the ROCStories dataset, we find that FiLM, when fine-tuned from the bidirectional RoBERTa-large, achieves a lower loss (2.63) compared to when fine-tuned from the unidirectional GPT2-xl (2.90), despite the former having only a quarter of the parameters of the latter. Therefore, we choose RoBERTa-large as the base model for FiLM here. We compare FiLM against

| Model | ROUGE-1/2/L |
|---|---|
| CM-1.6B | 33.15/9.03/31.94 |
| FiLM-0.4B | **38.19/13.24/36.49** |

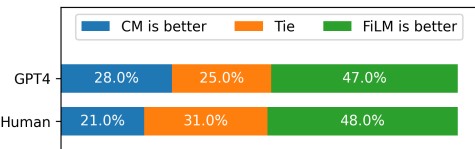

Figure 8: Story completion results on ROCStories. The left table presents the ROUGE scores, and the right figure illustrates the comparative evaluation by both GPT4 and human judges.

| 2. CM | I tried going to the park the other day. The weather seemed nice enough for a walk. I walked for a few hours. My eyes were watery and it was hard to breathe. My allergies were too bad and I had to go back home. |
|---|---|
| 1. FiLM | I tried going to the park the other day. The weather seemed nice enough for a walk. However, when I got there I started to itch. My eyes were watery and it was hard to breathe. My allergies were too bad and I had to go back home. |
| GPT4 | Option 1 introduces a sudden onset of the narrator's allergies, implicating that the park's environment may be causing it. This makes sense as it leads directly to the narrator's symptoms and decision to return home. Option 2 introduces a time lapse - the narrator's long walk - but this clashes with the sudden onset of the symptoms. Hence, option 1 maintains the logical flow of the story and makes the most sense in this context. Therefore, my final decision is <1>. |
| 2. CM | Josh loved when his mom baked apple pie. She always made it the same way. So he decided this time he would sneak a piece before dinner. The eggs his mom used must have been bad though. Josh got sick. |
| 1. FiLM | Josh loved when his mom baked apple pie. But he didn't like to eat it until it was ready. So he decided this time he would sneak a piece before dinner. The eggs his mom used must have been bad though. Josh got sick. |
| GPT4 | Option 1 suggests that Josh doesn't usually eat the apple pie before dinner because he prefers it when it's properly done, hence he deviated from his norm which resulted in him getting sick. This setup lends more of a cause-and-effect relationship between him deciding to sneak a piece early and getting sick, keeping the connection between story events clearer. Option 2 points out Josh's mom's consistency in making the apple pie but doesn't adequately explain why he chose to sneak a piece before dinner this time. It also doesn't tie as effectively to the ending where Josh gets sick. In terms of maintaining a logical flow and coherence, Option 1 seems to do a better job. Final decision: <1>. |

Figure 9: Story completion examples from ROCStories, evaluated by GPT4. The provided context is in black, and the model generated text is in color. The number before each model denotes the option number presented to GPT4 for evaluation.

the causal masking model (CM) fine-tuned from GPT2-xl, using top-p sampling with a threshold of 0.95 and a temperature of 0.2 for decoding.

The results depicted in Fig. 8 indicate that FiLM outperforms CM by an average of 4.6 ROUGE score points. Moreover, FiLM is preferred by both GPT4 and human evaluators, with preference margins of 19% and 27%, respectively. Fig. 9 showcases several story completions. While CM's generated sentence aligns with the prior context, it struggles to link appropriately with the sentences that follow. In contrast, FiLM's outputs exhibit seamless integration, like mentioning "started to itch" considering the subsequent allergy, and adding "But he didn't like to eat it until it was ready" before "So he decided this time he would sneak a piece before dinner", thereby preserving the story's logical flow. For more examples illustrating their qualitative difference, please refer to Fig. 12 in the Appendix.

## 5 CONCLUSION

In this paper, we have introduced FiLM, a novel language modeling approach capable of flexibly generating output sequences in any order. Analogous to MLMs, FiLM is trained to predict masked tokens conditioning on the surrounding context from both sides. Its masking probabilities are randomly drawn from a Beta distribution, allowing FiLM to generate sequences from scratch.

We have devised a method to compute the perplexity of any-order language models, including FiLM, to facilitate direct comparisons with CLM. In language modeling, FiLM's perplexity performance is competitive, nearing that of comparable CLMs as the models scale up in size. This suggests FiLM's promising potential as a new bidirectional LLM. FiLM excels in text infilling and story completion tasks, where it outperforms strong baselines in terms of both automatic and human evaluations.

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

# APPENDIX

| 2. CM | WASHINGTON ( Reuters ) - The United States has been secretly detaining an Iraqi held by the U.S. military, a British legal charity charged on Monday, but the Pentagon described the report as inaccurate. |
|---|---|
| 1. FiLM | WASHINGTON ( Reuters ) - The United States has been secretly detaining terror suspects for longer than previously reported, a leading British legal charity charged on Monday, but the Pentagon described the report as inaccurate. |
| GPT4 | Option 1 seems to refer to the prolonged detention of multiple terror suspects, while option 2 specifies the detainment of a particular Iraqi individual by the U.S military. Both options could potentially fit depending on the broader context, but without further information, option 1 seems slightly more coherent as it aligns better with the criticism made by the British legal charity and the Pentagon's response to that criticism. Hence, I conclude with <1>. |
| 1. CM | The Yankees needed three games to beat Cleveland in last year 's AL opener this year and then won Game 1 against the Indians. |
| 2. FiLM | The Red Sox came from behind to beat the Tigers in the American League 's Series opener this year and then won Game 1 against the Indians. |
| GPT4 | Both options are grammatically correct. However, they both present different scenarios. Choice 1 states that "The Yankees" needed three games to defeat Cleveland in the AL opener that happened last year, and then it leaps to discuss a victory in Game 1 against the Indians this year. This could potentially cause confusion due to the time jump in the sports events it discusses. Choice 2 says, "The Red Sox came from behind", implying that they played catch-up in a game to defeat the Tigers in the Series opener this year. They then, presumably in the same year, won a game against the Indians. This is more logically coherent and consistent in regard to its timeline. So, option 2 is the better choice. Final decision: <2>. |
| 2. CM | The satellite weighed some 660 pounds, Israeli Haaretz newspaper reported, citing company officials. |
| 1. FiLM | The satellite weighs some 660 pounds, the Israeli Haaretz newspaper reported, citing defense company officials. |
| GPT4 | Both options are grammatically correct and coherent, however, the difference lies in the tense used. Option 1 uses the present tense 'weighs' suggesting the satellite still exists in its described state, while option 2 uses the past tense 'weighed' implying the satellite may no longer exist or have changed in some way. The best choice largely depends on the overall context and temporal setting of the sentence. Given that there's insufficient context to determine the most appropriate tense, both options are equally appropriate. For this reason, my final decision is <->. |
| 2. CM | As North Korea's close neighbor, traditional ally and major provider of economic aid, China is widely believed to hold the key to solving the North Korea conundrum. |
| 1. FiLM | As North Korea's close neighbor, traditional military ally and provider of economic aid, China is widely believed to hold the key to solving the North Korea conundrum. |
| GPT4 | Both sentences are grammatically correct, and both retain the intended meaning of the sentence. However, option 1 specifies that China is a "military" ally while option 2 does not specify the type of ally but states it's a "major" provider of economic aid. Both are fitting and coherent with the context. However, depending on the specific context, it may be more accurate to emphasize the military alliance (option 1) or the significant economic aid (option 2). Therefore, without further context, I conclude my decision with <->. |
| 2. CM | Puerto Rico's unemployment rate hit 14.8 % in January, according to the U.S. Bureau of Labor Statistics. |
| 1. FiLM | Puerto Rico had a record low rate of employment in the last year, according to the Bureau of Labor Statistics. |
| GPT4 | Both options are grammatically correct and fit sensibly into the context. However, option 2 provides a more detailed, specific statistic (14.8% unemployment in January) while option 1 gives a general statement that could be improved with more precise figures. Therefore, based on coherency and information value of the statement, option 2 is a better fit. Final decision: <2>. |

Figure 10: Additional text infilling examples from One Billion Word.

| | |
|---|---|
| 1. CM | = Kiss You ( One Direction song ) = " Kiss You " is a song recorded by English @-@ Irish boy band One Direction for their second studio album, Take Me Home ( 2012 ). It was released as the record's second single in Germany and the third overall single on 7 January 2013. The song was composed by Kristoffer Fogelmark, Kristian Lundin, Albin Nedler, Savan Kotecha, Shellback and its producers, Carl Falk and Rami Yacoub. " Kiss You " is an upbeat power pop song with electronic effects ; the lyrics detail a protagonist's infatuation with a significant other. Critics praised the song for its production, calling it a stand @-@ out track on Take Me Home. The track became the group's sixth top @-@ ten hit in Ireland and the United Kingdom, while attaining top @-@ forty positions in both Belgian territories ( Flanders and Wallonia ), as well as in Australia, Canada, Denmark, France, New Zealand, and the Netherlands. The single peaked at number 46 on the US Billboard Hot 100 and has been certified gold by the Recording Industry Association of America ( RIAA ) for shipments of 500 @,@ 000 copies. One Direction performed " Kiss You " on both the Vevo channel and during their 2013 Take Me Home Tour. In 2013, the song was performed on the first leg of their 2013 @-@ 2014 Take Me Home Tour. = = Background and release = = After releasing their debut album, Up All Night ( 2011 ), One Direction embarked on a promotional campaign for the album, including several music video releases and a series of live appearances. They released the lead single, " What Makes You Beautiful ", on 3 December 2011. A music video for " Kiss You ", directed by Anthony Mandler, was filmed for the single and its accompanying music video. The music video for " Kiss You " was released on YouTube on 5 December, and featured cameo appearances by Liam Payne, Leona Lewis, Alesha Dixon, and other music videos. The clip depicts the band shooting various scenes via a green screen, which include sequences reminiscent of iconic music videos, such as the ones for the Beatles'" You Never Give Me Your Money ", the dancing game Just Dance 2014, and is also one of the select songs available on the demo version. Additionally, it is the final main track on the US edition of Now That's What I Call Music! 46. |
| 2. FiLM | = Kiss You ( One Direction song ) = " Kiss You " is a song recorded by English @-@ Irish boy band One Direction for their second studio album, Take Me Home ( 2012 ). It was released as the record's second single in Germany and the third overall single on 7 January 2013. The song was composed by Kristoffer Fogelmark, Kristian Lundin, Albin Nedler, Savan Kotecha, Shellback and its producers, Carl Falk and Rami Yacoub. " Kiss You " is an upbeat power pop song with electronic effects ; the lyrics detail a protagonist's infatuation with a significant other. Critics praised the song for its production, calling it a stand @-@ out track on Take Me Home. The track became the group's sixth top @-@ ten hit in Ireland and the United Kingdom, while attaining top @-@ forty positions in both Belgian territories ( Flanders and Wallonia ), as well as in Australia, Canada, Denmark, France, New Zealand, and the Netherlands. The single peaked at number 46 on the US Billboard Hot 100 and has been certified gold by the Recording Industry Association of America ( RIAA ) for shipments of 500 @,@ 000 copies. One Direction performed " Kiss You " on both the 2012 and 2013 versions of the 2012 MTV Video Music Awards and on the red carpet at the 2012 Royal Variety Performance. The group performed the track live on The X Factor in the United Kingdom. The accompanying music video was directed by Ed Bales and features a series of concept and extravagant filmmaking techniques, including film grain, motion blur and other filters reminiscent of other music videos. The clip depicts the band shooting various scenes via a green screen, which include sequences reminiscent of iconic music videos by Michael Jackson, Michael Jackson, Diana Ross and The Supremes. Upon its release, " Kiss You " charted in both the Irish and UK Singles Chart, reaching number one on the UK Singles Chart. The song has charted in the top twenty of Australia, Canada, the United States, Sweden and the United Kingdom. The song is featured on the soundtrack of the dancing game Just Dance 2014, and is also one of the select songs available on the demo version. Additionally, it is the final main track on the US edition of Now That's What I Call Music! 46. |
| GPT4 | Both options provide a smooth transition that follows the flow of the given text. However, option 2 shows a better understanding of the song and its relevance at events such as the 2012 MTV Video Music Awards and the 2012 Royal Variety Performance. It also correctly mentioned that the music video was directed by Ed Bales. In contrast, option 1 references confusing elements, like appearances by Leona Lewis and Alesha Dixon without further explanation of their roles. The suggestion "the Beatles'" You Never Give Me Your Money " in option 1 also feels awkward and is not clear in its relation to the narrative. Therefore, my final decision is <2>. |
| 2. CM | = = = Aircraft = = = Ise was briefly fitted with an aircraft flying deck during her reconstruction and was initially equipped with four floatplanes, but they were removed during the modernization and only one floatplane, a Nakajima E8N2 biplane, was installed in late 1936. The conversion was not completed until late 1937. The first Ise @-@ class battleships were fitted with the Type 3 floatplane MMF3 as their aircraft carrier escorts until the initial Nakajima E4N2 biplanes were replaced by Nakajima E8N2 biplanes in 1938. = = Conversion to aircraft carrier = = = The sinking of the British capital ships Prince of Wales and Repulse by Japanese land @-@ based aircraft on the night of 8 / 9 June 1942 and the loss of an estimated 39 aircraft to aircraft carrier aircraft during the Battle of the Eastern Solomons in May 1943 increased concerns about the ability of Japanese battleships to defeat enemy aircraft. The loss of several aircraft on 10 December 1941 led the IJN to realize that battleships could not operate in the face of enemy aircraft and required friendly air support to protect them. |
| 1. FiLM | = = = Aircraft = = = Ise was briefly fitted with an aircraft flying @-@ off deck equipped with Nakajima EMF3 biplanes. At the start of the war, two of the ships embarked a total of eleven Mitsubishi F1M2 fighters. In 1933, the ship embarked a total of twelve Mitsubishi A2N2 biplanes. Two years later, the ship embarked a total of 24 Mitsubishi A1M3 fighters and nine Mitsubishi F1M2 biplanes. In 1935, the ship was fitted with three more fighters and three more in 1938 - 39. The initial Nakajima E4N2 biplanes were replaced by Nakajima E8N2 biplanes in 1938. = = Conversion to aircraft carrier = = The sinking of the British capital ships Prince of Wales and Repulse by Japanese land @-@ based aircraft on 10 December 1941 led the IJN to realize that battleships could not operate in the face of enemy aircraft and required friendly air support to protect them. |
| GPT4 | The first option provides a more thorough and chronological history of the aircraft on the Ise, complete with the models of the biplanes and fighters and the timeline of when they were embarked. The mention of the Nakajima E4N2 biplanes being replaced by Nakajima E8N2 in 1938 also fits neatly with the earlier detail. In addition, the use of '@-@' instead of '/' in the word 'off-deck' and 'land-based' is more grammatically correct. However, in the second paragraph of option 1, the word 'aircraft carrier' is not enclosed by brackets correctly. The second option doesn't seem to provide a complete account of the history of aircraft on Ise, mentioning only a few floatplanes. It also refers to the change of aircraft in 1938, but less distinctly than option 1. The '@-@' symbol is improperly used as a substitute for a slash in phrases such as '8 / 9 June 1942'. The use of 'aircraft carrier' in brackets is semantically correct, but grammatically incorrect as it is not enclosed by brackets properly, similar to option 1. Given these considerations, option 1 is a clear candidate as it provides a more extensive history of aircraft on the Ise. Therefore, despite the bracket error which is a shared mistake in both options, I choose <1>. |

Figure 11: Text infilling examples from WikiText-103.

| 1. CM | Avery was married with children. She was tired of her boring life. One day, she decided to meet up with an old boyfriend from college. They had a great time together. Avery regretted what she did the next day. |
|---|---|
| 2. FiLM | Avery was married with children. She was tired of her boring life. One day, she decided to meet up with an old boyfriend from college. They had a lot of fun and ended up spending all night hanging out. Avery regretted what she did the next day. |
| GPT4 | Both options have similar themes and continue the story logically. Option 2 is slightly more detailed and specific about what occurred during the meet-up. It explicitly highlights the carefree time Avery and her old boyfriend had, which strongly reinforces her regret in the following sentence. Therefore, option 2 maintains the story's coherence slightly more than option 1. So, my final decision is <2>. |
| 2. CM | Feliciano went olive picking with his grandmother. While they picked, she told him stories of his ancestors. Before he realized it, the sun was going down. He had a nice dinner and went to bed. Feliciano was happy about his nice day. |
| 1. FiLM | Feliciano went olive picking with his grandmother. While they picked, she told him stories of his ancestors. Before he realized it, the sun was going down. He had a great time with his grandmother that day. Feliciano was happy about his nice day. |
| GPT4 | Option 1 keeps the focus of the story on the olive picking day and the relationship between Feliciano and his grandmother, thus adding to the coherence. Option 2, on the other hand, introduces a new event (dinner and going to bed) which does not contribute to the main plot and seems somewhat unrelated. Therefore, option 1 is more fitting. Final decision: <1>. |
| 1. CM | Kelly was playing her new Mario game. She was having a lot of fun. She was playing for so long without beating the level. Finally she beat the last level. Kelly was so happy to finally beat it. |
| 2. FiLM | Kelly was playing her new Mario game. She was trying to beat a level. She was playing for so long without beating the level. Finally she beat the last level. Kelly was so happy to finally beat it. |
| GPT4 | Both options seem to flow well with Kelly's actions throughout the story, filling in the missing part in a coherence manner. However, option 2 presents a more logical flow as it introduces the fact that Kelly was trying to beat a level. This ties in more directly with the remaining parts of the story where she is playing for so long without beating the level and finally manages to do it. Option 1 simply comments on her emotional state, which doesn't seem to link as directly to the following sentences. Therefore, option 2 seems to be the more fitting choice. Final decision: <2>. |
| 1. CM | Oliver was getting married. He was worried that he would stutter during the vows. When the time came, he took a deep breath and began to speak. He stuttered, but his wife smiled and hugged him and he was okay. Oliver was so grateful for his wife's love. |
| 2. FiLM | Oliver was getting married this weekend. He was worried that he would stutter during the vows. When the time came, he took a deep breath and began to speak. He stuttered, but his wife smiled and hugged him and he was okay. Oliver was so grateful for his wife's love. |
| GPT4 | Both options are very similar, and both provide continuum and coherence for the rest of the story. The difference between them is slight: option 1 states that Oliver was getting married, while option 2 specifies that the wedding was taking place this weekend. Both are logical and fit into the story seamlessly. Thus, neither is more fitting than the other. Answer: <->. |
| 1. CM | Brandon was given a 10 page research paper to write for class. He begun working on the paper on his 2003 Dell laptop. Brandon stayed up until 1 in the morning to try and finish his paper. Suddenly, his computer crashed and died. Brandon was so upset that he had to turn in his paper late. |
| 2. FiLM | Brandon was given a 10 page research paper to write for class. He begun working on the paper on his 2003 Dell laptop. Brandon stayed up until 1 in the morning to try and finish his paper. Suddenly, his computer crashed and died. Brandon was devastated that he would never finish his paper. |
| GPT4 | Option 1 suggests that Brandon had his paper completed, or nearly completed, but was unable to turn it in on time due to his computer crashing. This is a logical flow of events. Option 2 suggests that because of the computer crashing, Brandon was unable to finish his paper, but this does not necessarily mean he would never be able to finish it. Therefore, it seems less logical and coherent. Hence, option 1 is the most suitable choice. Therefore, the answer is <1>. |

Figure 12: Additional story completion examples from ROCStories.

Table 3: Agreement percentages between GPT4-human and human-human evaluations on ROCStories. Each cell shows the percentage where the row head matches the first evaluator's preference and the column head the second's. **Bold** entries highlight agreements, and underlined entries mark opposite preferences for options 1 and 2.

| GPT4\Human | 1 | 2 | Tie |
|---|---|---|---|
| **1** | **21.0** | 8.5 | 11.5 |
| **2** | 6.0 | **19.0** | 10.0 |
| **Tie** | 7.0 | 7.5 | **9.5** |

| Human\Human | 1 | 2 | Tie |
|---|---|---|---|
| **1** | **24.0** | 3.0 | 8.0 |
| **2** | 5.0 | **22.0** | 14.0 |
| **Tie** | 4.0 | 4.0 | **16.0** |

