# OpenReview forum: "FiLM: Fill-in Language Models for Any-Order Generation"
_ICLR.cc/2024/Conference — Submitted to ICLR 2024_

### Official Review · Reviewer_FCE8 · 2023-10-30

**Soundness:** 3 good
**Presentation:** 3 good
**Contribution:** 3 good
**Rating:** 5
**Confidence:** 4

**Summary:**

This paper proposes the Fill-in Language Model (FiLM), a new language modeling approach that allows for flexible generation at any position without adhering to a specific generation order. Its training extends the masked language modeling objective by adopting
varying mask probabilities sampled from the Beta distribution to enhance the generative capabilities of FiLM. During inference, FiLM can seamlessly insert missing phrases, sentences, or paragraphs, ensuring that the outputs are fluent and are coherent with the surrounding context. In both automatic and human evaluations, FiLM outperforms existing infilling methods that rely on left-to-right language models trained on rearranged text segments. FiLM is easy to implement and can be either trained from scratch or fine-tuned from a left-to-right language model.  The results are promising

**Strengths:**

* FiLM's flexibility to generate text in any order is a novel capability lacking in most LMs.
* The adaptive masking strategy for training is simple but impactful.
* Computing perplexity for non-causal LMs is an important contribution.
* Strong quantitative and qualitative results for text infilling and completion.
* Ablations clearly validate the design choices like Beta distribution masking.

**Weaknesses:**

* FiLM lags behind causal LMs in terms of perplexity, especially for smaller model sizes.
* Limited analysis of how perplexity varies across different decoding strategies.
* No exploration of other pretraining objectives tailored for any-order generation.

**Questions:**

NA

---

### Official Review · Reviewer_Ad3z · 2023-10-31

**Soundness:** 3 good
**Presentation:** 2 fair
**Contribution:** 3 good
**Rating:** 6
**Confidence:** 3

**Summary:**

This paper describes FiLM (Fill-in Language Model), a novel approach to non-causal language modeling, wherein the model predicts multiple masked tokens in an input sequence that can be variable in number and flexible in locations. FiLM is trained using a novel masking schedule based on a Beta distribution, which is shown in this paper to outperform the fixed masking probability and a uniform distribution in terms of achieving lower perplexities on eval datasets. FiLM can be trained on pretrained foundation models that are either masked LMs (MLMs) or causal LMs (CLMs). The authors conducted experiments with RoBERTa as an example of the MLMs and with GPT-2 as an example of CLMs. The authors show that FiLMs achieve substantially higher ROGUE scores (indicating higher quality) for the text infilling task compared to a causal masking (CM) baselines. On a causal language modeling benchmark, FiLM show worse (higher) perplexity compared to a traditional CLM, which was expected. But these perplexity gaps diminish with increasing model size. The authors also provide a few concrete examples of text filled in by FiLM and compared them with outputs of CM on the same input prompts, showing a clear advantage in terms of the coherence of the filled text by FiLM. Evaluation of infilling results based on human raters and GPT-4 show a clear overall preference for FiLM compared to CLMs.

**Strengths:**

S1. A clearly-written report, easy to follow. It describes the general motivation, methodology, and results of FiLM well.
S2. Introducing a novel noise schedule for masking tokens for the MLM training objective. Convincingly demonstrating its advantage over other noise schedule including the uniform distribution and a fixed mask probability.
S3. Defining a number of decoding strategies (schedules) for FiLM to accommodate multiple masks in a sequence, including random, left-to-right, right-to-left, min-entropy, and max-entropy. Using empirical results to illustrate the relative advantage of left-to-right and min-entropy.
S4. Using a diverse set of evaluation datasets including WikiText-103, 1BW, and ROCStories, employing both automated metrics such as ROGUE and subjective metrics such as pairwise preference (according to human raters or GPT4).

**Weaknesses:**

W1. The presentation in the manuscript lacks some details (see my questions below). But that should be addressable by the authors.
W2. Lack of evaluation on an out-of-domain evaluation set. The evaluation of FiLM that the authors conducted over the 1BW and ROCStories datasets are based on FiLMs trained on these datasets. Therefore it remains to be shown that the infilling capability of FiLM can be extended to text domains unseen during training. This feels important for the practical usefulness of FiLM in real-world applications.
W3. Relating to point W2 above, the author should better motivate FiLM and its infilling capability. In what real-world applications would FiLM be useful? Would FiLM be useful as a foundational model for non-infilling tasks such as classification?

**Questions:**

Q1. Need more information on human evaluation procedures. Who were the human raters? How were they recruited? What was the task like? How were they trained on this task? Was the mask part revealed to the human rater while this task is performed? How was inter-rater reliability (IRR) evaluated?
Q2. A question related to Q1 was how the GPT4 model was prompted to perform the rating task, including the prompt template, the sampling temperature, etc.
Q3. What tokenizer(s) did the RoBERTa and GPT-2 models use? For fair comparisons (e.g., results in Figure 5), they ought to be based on the same tokenizer. Some tokenizers are at the subword level. For example, is it possible that certain words are split into two parts and masked partially during mask generation? How does this affect the training, decoding, and human evaluation?
Q4. How do the training examples look like? What is the output shape of FiLM? Is it the set of probability scores over the vocabulary for a single token, or the sets of scores for multiple tokens. This is related to the question whether a sequence with N masks (where N can be >1 in general) yields a single or multiple (N) training examples.
Q5. There seems to be some discrepancy between Figures 6 and 7 (right panels). While Figure 6 Right Panel shows only the eval results from GPT4, Figure 7 Right Panels show both human and GPT4 rating results. Why not show human rating results in Figure 6 (i.e., for the WikiText-103 and 1BW datasets) as well?

---

### Official Review · Reviewer_Q9uj · 2023-11-01

**Soundness:** 2 fair
**Presentation:** 3 good
**Contribution:** 1 poor
**Rating:** 3
**Confidence:** 3

**Summary:**

The authors propose a train an MLM with a new masking ratio instead of a fixed 15% which is the standard in BERT. Specifically, they propose to draw a masking ratio from a beta distribution with a mode of 50% masking ratio.

**Strengths:**

1. Results show the proposed beta(2.5,2.5) masking ratio is better than uniform[0,1] and fixed ratios.

**Weaknesses:**

1. Table 1 shows the proposed beta(2.5,2.5) masking ratio is only marginally better than U[0,1] on two datasets.
2. The paper does not have enough content. For example, the authors use a lot of case studies (Figures 1,4, 7) to waste space.
3. The authors do not compare with non-autoregressive methods, for example (1), which is important because they both address the non-left-to-right generation problem.

(1) Insertion Transformer: http://proceedings.mlr.press/v97/stern19a/stern19a.pdf

**Questions:**

1. Since the improvement is not very significant (Table 1), have the authors verified how robust the improvement is? For example, running multiple rounds and finding the average and standard deviation.

2. How does the proposed beta masking ratio work with different alphas and betas other than (2.5,2.5)?

---

### Official Review · Reviewer_Bw7d · 2023-11-01

**Soundness:** 4 excellent
**Presentation:** 3 good
**Contribution:** 2 fair
**Rating:** 3
**Confidence:** 5

**Summary:**

Summary:

The paper introduces the Fill-in Language Model (FiLM), which enhances language models by allowing for flexible text generation in any order, thus overcoming the limitations of traditional left-to-right generation and making use of bidirectional context. FiLM's training employs varying mask probabilities drawn from the Beta distribution, optimizing its generative abilities. During inference, it can insert text at any point, maintaining fluency and contextual coherence. FiLM can be trained from scratch or fine-tuned from existing models, showing scalability and potential as a large language model with its perplexity nearing that of conventional models as size increases. In tasks such as text infilling and story completion, FiLM outperforms existing methods, as confirmed by both automatic and human evaluations.

**Strengths:**

Advantages:

 - Flexible Text Generation: FiLM allows for generating text in any desired order, not just left-to-right, leveraging bidirectional context which is beneficial for tasks involving text infilling and editing.
 - Adaptive Training Strategy: The model uses varying mask probabilities sampled from the Beta distribution during training, enhancing its generative capabilities and allowing it to adaptively learn from different contexts.
 - Scalability and Performance: It exhibits competitive perplexity compared to causal language models (CLMs), especially as it scales up in size, and it outperforms strong baselines in text infilling and story completion tasks, according to both automatic and human evaluations​.

**Weaknesses:**

Disadvantages:

 - Unfair Comparison: Correct me if I'm wrong but it seems to me the comparison in the infilling task can be unfair. The formulation of FiLM implicitly assumed the total number of infilled tokens, whereas this is not the information that CM-based infilling models know. You may have observed that the output from CM-based infilling models usually are longer compared to FiLM's. This could cause extra difficulties for CM-based infilling and eventually make the comparison unfair and less meaningful.
 - Training/Inference Discrepancy: It seems to me while the proposed mask scheduler can alleviate it, there's still a training/inference behaviorial inconsistency in the proposed algorithm. I would appreciate it if some ablation study can be conducted to show how effective the advanced beta-Distribution-guided masking is and how severe is this discrepancy.
 - Under Explored Backgrounds: There are many preliminary works that seem relevant but not discussed or even mentioned in the paper. InDIGO [https://arxiv.org/abs/1902.01370], for example, is one of the first attempts at generating text in arbitrary order using insertion-based models. It even has an advantage against the proposed approach that it does not assume the length of span of the infilling. Following InDIGO there are also other insertion-based models that can achieve the same goal and can be trained efficiently, like InsNet [https://arxiv.org/abs/2102.11008].

**Questions:**

Please refer to Weakness).

---

### Meta-Review · Area_Chair_BYp5 · 2023-12-09

**Metareview:**

The paper introduces the Fill-in Language Model (FiLM), which enhances language models by allowing for flexible text generation in any order, thus overcoming the limitations of traditional left-to-right generation and making use of bidirectional context.

There are several concerns about the empirical settings, clarity, analysis, etc. Therefore, this paper needs significant improvements before acceptance.

**Justification For Why Not Higher Score:**

There are several concerns about the empirical settings, clarity, analysis, etc. Therefore, this paper needs significant improvements before acceptance.

**Justification For Why Not Lower Score:**

N/A

---

### Decision · Program_Chairs · 2024-01-16

Reject